# Crossover from gas-like to liquid-like molecular diffusion in a simple supercritical fluid

Umbertoluca Ranieri [1,2], Ferdinando Formisano [3,4] ✉,
Federico A. Gorelli [5,6,7] ✉, Mario Santoro [7,8], Michael Marek Koza [4],
Alessio De Francesco [3,4] & Livia E. Bove [1,9,10]

According to textbooks, no physical observable can be discerned allowing to distinguish a liquid from a gas beyond the critical point. Yet, several proposals have been put forward challenging this view and various transition boundaries between a gas-like and a liquid-like behaviour, including the so-called Widom and Frenkel lines, and percolation line, have been suggested to delineate the supercritical state space. Here we report observation of a crossover from gas-like (Gaussian) to liquid-like (Lorentzian) self-dynamic structure factor by incoherent quasi-elastic neutron scattering measurements on supercritical fluid methane as a function of pressure, along the 200 K isotherm. The molecular self-diffusion coefficient was derived from the best Gaussian (at low pressures) or Lorentzian (at high pressures) fits to the neutron spectra. The Gaussian-to-Lorentzian crossover is progressive and takes place at about the Widom line intercept (59 bar). At considerably higher pressures, a liquid-like jump diffusion mechanism properly describes the supercritical fluid on both sides of the Frenkel line. The present observation of a gas-like to liquid-like crossover in the self dynamics of a simple supercritical fluid confirms emerging views on the unexpectedly complex physics of the supercritical state, and could have planet-wide implications and possible industrial applications in green chemistry.

Matter can be pushed to temperatures and pressures above those of its critical point, in a state, called supercritical fluid (SCF), where we are unable to distinguish whether the system is a liquid or a gas. As a result, supercritical fluids possess an intimate hybrid nature: they are said to have the capability to effuse like a gas but to dissolve materials like a liquid. This makes them used in the most disparate industrial processes, such as pharmaceutical and cosmetics processing, extracting essential oils, dissolving waste material, brewing, or decaffeinating coffee beans without leaving toxic residues. In particular, near the critical point, small changes in pressure or temperature result in large

[1]Dipartimento di Fisica, Università di Roma La Sapienza, Piazzale Aldo Moro 5, Roma 00187, Italy. [2]Centre for Science at Extreme Conditions and School of Physics and Astronomy, University of Edinburgh, Edinburgh EH9 3FD, UK. [3]CNR - Istituto Officina dei Materiali (IOM), Grenoble, INSIDE@ILL, 71 Avenue des Martyrs, Grenoble Cedex 9, France. [4]Institut Laue-Langevin, 71 Avenue des Martyrs, Grenoble Cedex 9, France. [5]Center for High Pressure Science and Technology Advanced Research (HPSTAR), 1690 Cailun Road, Shanghai 201203, China. [6]Shanghai Advanced Research in Physical Sciences (SHARPS), Pudong, Shanghai 201203, China. [7]Consiglio Nazionale delle Ricerche, Istituto Nazionale di Ottica, CNR-INO, Via Nello Carrara 1, Sesto Fiorentino (FI) 50019, Italy. [8]European Laboratory for Nonlinear Spectroscopy, LENS, Via Nello Carrara 1, Sesto Fiorentino (FI) 50019, Italy. [9]Laboratory of Quantum Magnetism, Institute of Physics, École Polytechnique Fédérale de Lausanne, Lausanne CH-1015, Switzerland. [10]Institut de Minéralogie, de Physique des Matériaux et de Cosmochimie, Sorbonne Université, UMR CNRS 7590, 5 Place Jussieu, Paris 75005, France. ✉e-mail: formisano@ill.fr; federico.gorelli@hpstar.ac.cn

changes in density, allowing many properties of a supercritical fluid to be fine-tuned in such processes[1–6].

Understanding the physical behaviour of supercritical fluids is not only important for industrial applications but also to gain a much better knowledge of what goes on deep inside giant planets like Jupiter, Saturn, Uranus or Neptune. As we delve into their atmosphere, we may indeed encounter simple matter in the supercritical fluid state[7,8]. This will determine not only their thermal properties, and in particular the way heat is stored and flows around planets, but also the anomalous solubility of other volatiles in their atmosphere[9,10], as well as other surprising phenomena such as the separation of Uranus atmosphere from its interior, recently suggested by the anomalously low heat flow measurements from the Voyager spacecraft[11]. Considering the number of exoplanets and super-Earths that are being discovered, many of them larger and with more extreme environments than found in our own Solar System, the possibility for supercritical fluids to be found in planetary environments is only set to increase. Furthermore, SCFs with gas-like viscosity as well as melt-like wetting and element-carrying capability are an ideal agent for chemical transport at subduction zones and could play a role as life-sustaining solvents on other worlds[12,13].

Above the critical temperature, no increase in pressure can liquefy the system and when the pressure is greatly increased, a supercritical fluid of much higher density, close to the one of a solid, is produced. The significant changes in density observed close to the critical point are expected to correlate with changes in other properties of the fluid. The question scientists have long debated is: may a region of the pressure-temperature (*P-T*) plane and a physical observable be discerned allowing to unambiguously distinguish between gas-like and liquid-like behaviour in the supercritical fluid? As an answer to this question, various transition boundaries have been suggested to delineate the supercritical state space, including the Widom line, the Frenkel line, and the percolation line.

The Widom line, i.e. the line emanating from the critical point which represents the maxima/minima in certain thermophysical properties, has been suggested to separate liquid-like behaviour and gas-like behaviour in supercritical fluids. A crossover in the anomalous dispersion (i.e. the difference between the unrelaxed and relaxed value) of the simulated THz sound velocity[14–16], the observation of a pseudo-boiling phenomenon[17,18] and of droplet formation[19], and a change in density fluctuations correlations[20,21] have been reported to occur when crossing the Widom line. Another scenario envisages the Frenkel line as the line separating a gas-like (non-rigid) diffusive behaviour of the fluid from a rigid one at higher pressures. This is defined as the line along which the structural relaxation time equals the shortest period of transverse oscillations of the particles (atoms or molecules) within the system, and has been extensively investigated experimentally[22–24] and computationally[25–28]. On the high-pressure side of the Frenkel line, with the exception that the particles occasionally hop from one position to another, dense SCFs behave more like solids. They can sustain transverse acoustic excitations and show a velocity autocorrelation function which monotonically decays to zero. The percolation line[29] is directly rooted in microscopic structure and designates the critical density at which molecular clusters aggregate and shift from micro- to macroscopic dimensions. Percolation theory[30] was applied to identify crossovers in various supercritical substances, including argon, carbon dioxide, and water. The percolation crossover does not coincide with but closely follows the Widom line[31], and was verified experimentally in supercritical water through neutron diffraction[32].

Recent molecular simulations[33–35] and statistical mechanics studies[36–38] of the microscopic structure of supercritical fluids have pointed at the existence of localized density fluctuations in supercritical fluids, also marking a transition between the gas-like and the liquid-like state. These density fluctuations stem from molecular clusters interspersed among unbound molecules, with each cluster comprising fluid molecules that enhance the local density. Various investigations have explored the structural, thermodynamic, and dynamic aspects related to this crossover condition, confirming its existence experimentally in supercritical water[17,18] and in supercritical carbon dioxide[19]. Finally, studies measuring and simulating sound dispersion in high-pressure argon[14] and investigating longitudinal current correlation spectra in supercritical water[39] have provided insights into the transition across the Widom line, suggesting that this line, rather than the Frenkel line, compartmentalizes the supercritical state space.

Here we report a quasi-elastic neutron scattering (QENS) study of supercritical methane as a function of pressure at a constant temperature of 200 K (just above the critical temperature of 190.55 K), providing a direct measurement of the molecular diffusion in the system on the picosecond time and Å length scales. We find that the measured self-dynamic structure factor, i.e. the space and time Fourier transform of the self pair correlation function[40], shows a gradual change from a Gaussian lineshape, describing the molecular diffusion in a gas, to a Lorentzian lineshape, describing the molecular diffusion in a liquid. The change is completed when pressure crosses the Widom line and no further changes could be detected at higher pressures, for example when crossing the Frenkel line. The molecular diffusion coefficient has been determined over a range of densities covering almost three orders of magnitude and compared to both (non-ideal) gas and dense-fluid theoretical descriptions. The observation of this crossover in supercritical methane could have planet-wide implications, in particular due to the ubiquitous presence of methane in our Solar System. In gas giants, the existence of a non-homogeneous supercritical state would indeed influence whether a boundary between the planets' interior and their atmosphere can be defined. This would impact planetary properties such as the thermal conductivity and other heat-related physical phenomena like storm activity, as well as mass diffusion-related properties like the ionic conductivities and the consequent generation of anomalous magnetic fields[41–43].

## Results

### Investigated part of the phase diagram and measured neutron spectra

Figure 1 reports the temperature-pressure phase diagram of methane. The critical point is at $T_C$=190.55 K and $P_C$= 46 bar. Data at seventeen pressures between 8.5 and 2450 bar were acquired for this study along the 200 K isotherm (corresponding to 1.05 $T_C$), six below and eleven above $P_C$. The used high-pressure setup is described in the Methods section. The investigated pressure range corresponds to a very substantial density variation, from 0.00868 g cm$^{-3}$ at 8.5 bar to 0.47101 g cm$^{-3}$ at 2450 bar (5000% of the initial value). The first value is only an order of magnitude higher than the density of gas methane at ambient pressure (0.00097 g cm$^{-3}$ at 200 K). The second value is comparable to the density of liquid methane (for example, 0.4389 g cm$^{-3}$ at ambient pressure and 100 K). Density values used throughout this paper are taken from the equation of state of the National Institute of Standards and Technology (NIST) Chemistry WebBook, which uses ref. 44.

The pressure dependence of the density along the 200 K isotherm is also reported in Fig. 1 (upper panel). The intercepts of the Widom line and of the Frenkel line with the 200 K isotherm are at 59[44] and 670 bar[27], respectively. The criterion chosen to locate the Widom line is the locus of the isobaric heat capacity maxima; choice of a different thermophysical property would not significantly affect the intercept with the 200 K isotherm. The criterion chosen to locate the Frenkel line in ref. 27 is the disappearance of the first minimum in the calculated velocity autocorrelation function. Choice of a different signature such as heat capacity, saturation of the coordination number, or minimum in the Raman frequency might provide a considerably different result[45]. However, no other studies tracking the Frenkel line position

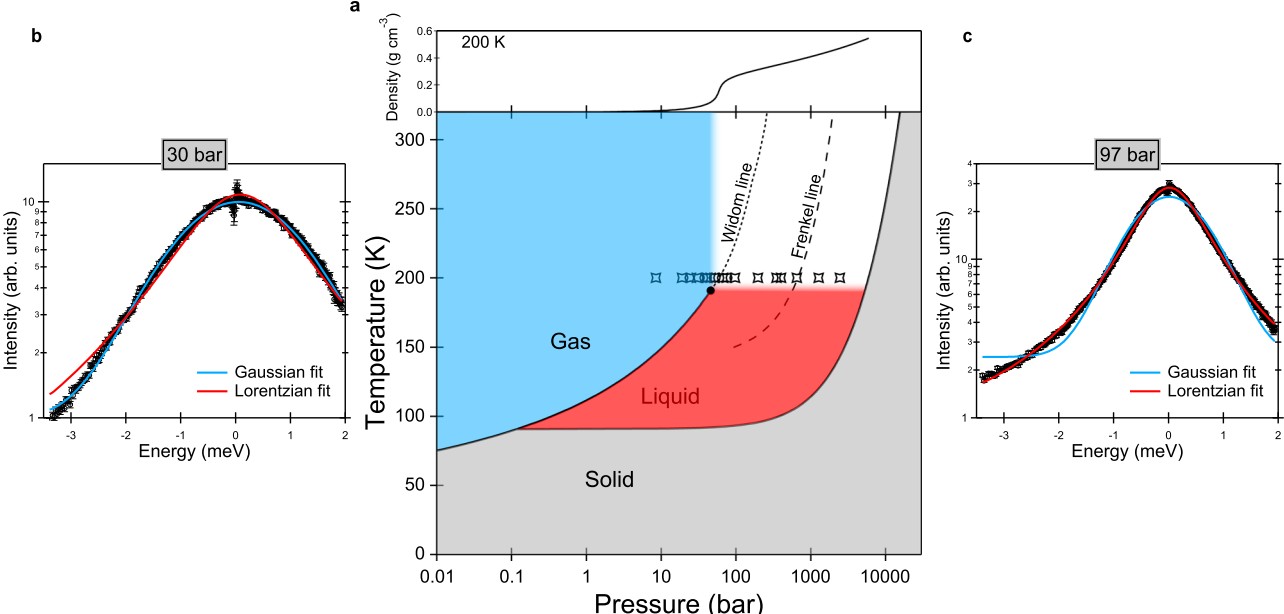

**Fig. 1 | Phase diagram of methane. a** Temperature-pressure phase diagram of methane including the Widom line (locus of the isobaric heat capacity maxima[44]) and the predicted Frenkel line from ref. [27]. The critical point is at 190.55 K and 46 bar. Symbols depict the pressure points measured (along the 200 K isotherm) in the present experiment. The density profile at 200 K from ref. [44] is reported on top of the phase diagram. **b**, **c** Examples of QENS experimental spectra in logarithmic scale (empty circles) at two selected pressures on each side of the Widom line with their best Gaussian/Lorentzian fits (solid lines), highlighting the Gaussian-to-Lorentzian crossover upon compression. To improve statistics, these two spectra were obtained by summing up measurements in the 0.55–0.85 Å⁻¹ exchanged wavevector range. Error bars were calculated by the square root of absolute neutron count combined with the law of propagation of errors.

on the phase diagram of methane are available in the literature to our knowledge. Figure 1 also anticipates the main result of this study: We observe a crossover in the lineshape of the experimental spectra (proportional to the self-dynamic structure factor of the system) from Gaussian to Lorentzian upon pressure increase (see the two examples of measured neutron spectra and their fits in Fig. 1), which is completed above the Widom line intercept.

Figure 2 depicts examples of QENS spectra of the sample, plotted as a function of the energy transfer $E$, for a few selected $P$ and $Q$ values, $Q$ being the modulus of the wavevector transfer. It is well-known that because of the very large incoherent neutron scattering cross section of ¹H, the QENS signal from $CH_4$ is dominated by the *self* (single-particle) dynamics of the molecules. In particular, the *translational* self-dynamic structure factor was measured in the present study. The measured (quasi-elastic) signal monotonically gets broader either with decreasing pressure at constant $Q$ (see Supplementary Fig. 1) or with increasing $Q$ at constant $P$ (see below), as expected in case of translational motion. It can be remarked that, at low pressures, the effect of pressure on the spectra is very small. As can be seen in Fig. 2, spectra are best fitted by a Gaussian function at low pressures and by a Lorentzian function at high pressures. The Gaussian-to-Lorentzian transition is progressive. At about 40–60 bar, the Gaussian and Lorentzian fits to our spectra are of comparable quality and none of them perfectly reproduces the lineshape of the experimental spectra (see Fig. 2). Gaussian fits better describe the data at all pressures up to 51.5 bar while Lorentzian fits are more adequate at and above 67.5 bar. More details about this are given below. More details about the extraction of spectra from the raw data and about the fitting of the spectra are given in the Methods section.

### Data analysis

In dilute fluids, the frequency of the collisions between particles is low and the particles experience a ballistic motion between two collisions. As a consequence, a velocity and a trajectory can be defined and the

mean square displacement (MSD) of the particle displays a $t^2$ time dependence. This behaviour translates, when switching to the experimental domain $(Q, \omega)$, into the free-particles (ideal gas) expression for the self-dynamic structure factor $S_{self}(Q, \omega)$ given by[40]:

$$S_{\text{self}}(Q,\omega) = \left(\frac{m}{2\pi k_B T Q^2}\right)^{1/2} \exp\left(-\frac{m\omega^2}{2k_B T Q^2}\right), \qquad (1)$$

which is a Gaussian function of the frequency $\omega$ (with $\omega = E/\hbar$) centered at $\omega = 0$, whose expression is independent on density and whose width increases linearly with $Q$. Here, $m$ is the mass of the particle and $k_B$ the Boltzmann constant. $2k_B T/m$ gives the mean square velocity of the diffusing particles. On the other hand, the self-diffusion coefficient $D$ is given by the kinetic Chapman-Enskog theory for a dilute fluid composed of hard spheres and is inversely proportional to the number density $n$ of the system. It can be approximated for the non-dilute hard-sphere fluid by[46–48]:

$$D = \frac{1.01896}{g(\sigma)} \frac{3}{8n\sigma^2} \left(\frac{k_B T}{\pi m}\right)^{1/2}, \qquad (2)$$

with the numerical factor 1.01896 being a result of using the ninth Enskog approximation, $\sigma$ being the spheres diameter, and $g(\sigma)$ being the radial distribution function at contact. Eq. (2) is almost exact in the low-density limit while its applicability at higher densities has been thoroughly tested in molecular dynamics calculations[47]. For the range over which Gaussian fits were employed in the present study, i.e. for a packing fraction $\zeta = \pi n\sigma^3/6$ up to about 0.1, Eq. (2) is accurate within 10% compared to recent computational values for hard spheres[49,50]. Moreover, there is extensive literature[51,52] showing that the hard-sphere model correctly reproduces the experimental diffusion coefficient of methane at such modest pressures, provided that the equivalent hard-sphere diameter is a free-fitting parameter. It follows that the free-particles expression of Eq. (1) can be extended at higher

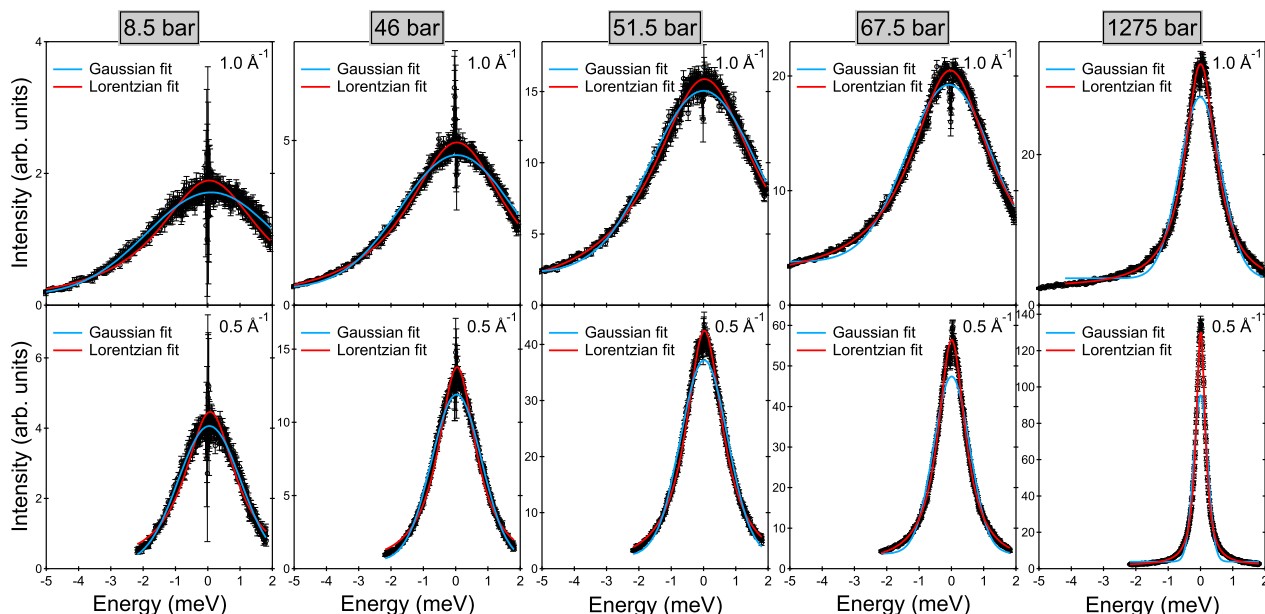

**Fig. 2 | Examples of QENS spectra.** QENS spectra of the sample at 200 K at the indicated pressure and wavevector transfer values, collected using an incoming neutron wavelength of 5.12 Å and an instrumental resolution of full width at half maximum equal to 0.077 meV. Experimental data (empty circles) are compared to the best Gaussian and Lorentzian fits (solid lines). Error bars were calculated by the square root of absolute neutron count combined with the law of propagation of errors.

densities using Eq. (2) and the Gaussian $S_{\text{self}}(Q,\omega)$ writes now as:

$$S_{\text{self}}(Q,\omega) = \left(\frac{\ln(2)}{\pi\Gamma_G^2(Q)}\right)^{1/2} \exp\left(-\frac{\ln(2)\omega^2}{\Gamma_G^2(Q)}\right), \tag{3}$$

$$\text{where} \quad \Gamma_G(Q) = KQ. \tag{4}$$

The values of the half width at half maximum $\Gamma_G(Q)$ obtained from the Gaussian fits of our spectra display a linear trend with $Q$, as shown in Fig. 3a, where the data are reported together with their linear fit. From the values of the fitted slope $K$, reported in Supplementary Fig. 2 at each investigated gas-like pressure, we estimated $D$ from the expression obtained by comparing Eqs. (1) to (4): $K = \frac{8}{3}\frac{(2\pi\ln(2))^{1/2}}{1.01896} n\sigma^2 g(\sigma)D$, using $g(\sigma) = \frac{1-\frac{\zeta}{2}}{(1-\zeta)^3}$ as given by the widely employed Carnahan-Starling equation of state[53] and $\sigma = 3.61$ Å as given for methane at 200 K by NMR measurements of $D$[52]. The derived self-diffusion coefficient is plotted in Fig. 4 as a function of pressure. It is an important and intrinsic peculiarity of our approach that literature density values are needed for the determination of $D$ in addition to the Gaussian widths. Using a macroscopic average for the density is certainly a rough approximation for a system that is possibly heterogeneous on the experimentally probed length scale. The used density values (from the NIST Chemistry WebBook) are tabulated in Supplementary Table 1.

When the density further increases and becomes comparable to the one of a liquid, the collision frequency also strongly increases and the particles experience several multiple uncorrelated collisions during the typical experimental time scales, giving rise to a random Brownian motion characterized by a linear, and not anymore quadratic, time dependence for the MSD[54,55]. As a consequence, the self-dynamic structure factor has a Lorentzian shape centered at $\omega = 0$, i.e.:

$$S_{\text{self}}(Q,\omega) = \frac{1}{\pi}\frac{\Gamma_L(Q)}{\omega^2 + \Gamma_L^2(Q)}, \tag{5}$$

which is the model customarily used to analyse QENS data on bulk liquid samples. Typically, the half width at half maximum $\Gamma_L(Q)$ is observed to be proportional to $Q^2$ at small $Q$ (as expected for Fickian diffusion) and to lie below that expectation at high $Q$, i.e., when one is probing shorter distances and the microscopic details of the motion become relevant. It is often correctly described by the formula:

$$\Gamma_L(Q) = \frac{DQ^2}{1 + \tau DQ^2}. \tag{6}$$

Eqs. (5) and (6) were previously employed in many QENS studies, including studies of liquid methane[56], supercritical methane at liquid densities[57], and high-pressure interfacial methane[58], as well as studies of other dense fluids such as for example liquid water over wide ranges of temperature and pressure[59,60]. Eq. (6) is known to be associated to the jump-like nature of the translational motion in dense systems, typically because of the presence of a cage effect, and $\tau$ represents the time spent by the diffusing particle at quasi-equilibrium sites between rapid jumps[61]. Figure 3b reports the half width of the Lorentzian fits of our spectra as a function of $Q$ at each investigated liquid-like pressure as well as its best fits using Eq. (6), from which we extracted values of $\tau$ and $D$, reported in Fig. 4.

## Obtained self-diffusion coefficient and comparison with the literature

As can be seen in Fig. 4, $\tau$ shows a 20% decrease with increasing pressure attaining a minimum at around the Frenkel line intercept (670 bar) followed by a small 10% increase above that pressure. However, the estimation of the residence time $\tau$ is dominated by the high-$Q$ values of $\Gamma_L(Q)$; its uncertainty can be significant for different reasons, such as a high background. Figure 4 also plots the obtained self-diffusion coefficient $D$ as a function of the pressure (merging together the gas-like Gaussian and liquid-like Lorentzian fit results); values range from $1481.1 \times 10^{-9}$ m²s⁻¹ at 8.5 bar to $6.8 \times 10^{-9}$ m²s⁻¹ at 2450 bar. At our highest pressure, the value of $D$ is close to the typical values of liquid methane (for example, $3.6 \times 10^{-9}$ m²s⁻¹ at 1.5 bar and 102 K[56]). In Fig. 4, the diffusion coefficient is compared with NMR data of methane

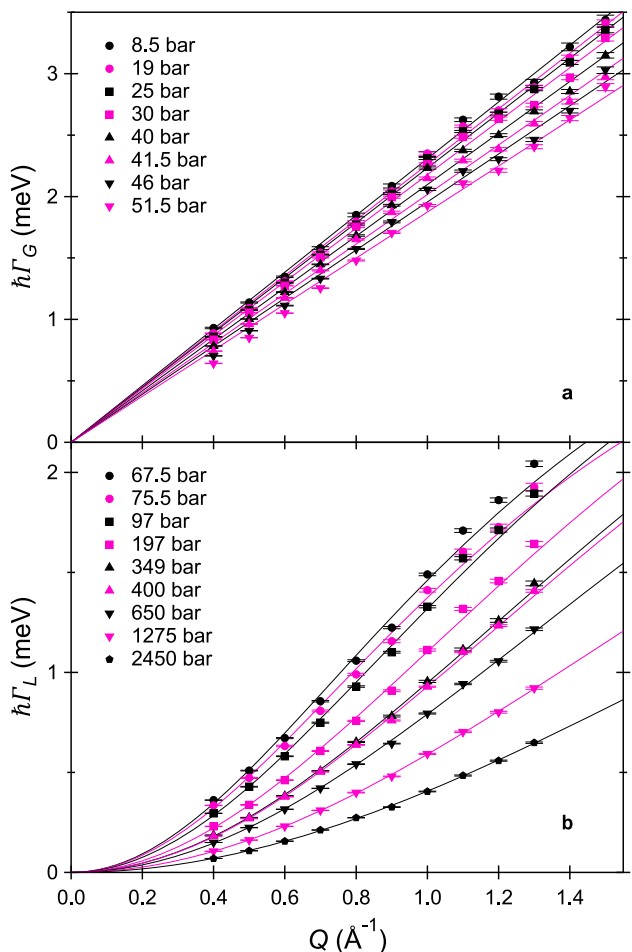

**Fig. 3 | Wavevector transfer dependence of $\Gamma_G$ and $\Gamma_L$.** Fitted Gaussian (**a**) or Lorentzian (**b**) half widths at half maximum (symbols) as a function of $Q$, at all investigated pressures. Error bars correspond to one standard deviation, as obtained from fits such as those in Fig. 2. The best fits to the data at each pressure are shown as solid lines. For the Gaussian half widths, the fits are lines passing through zero whose slopes allowed us to extract $D$ (see Eq. (4) and the relative discussion). The Lorentzian half widths were modelled using Eq. (6) with $\tau$ and $D$ as free parameters.

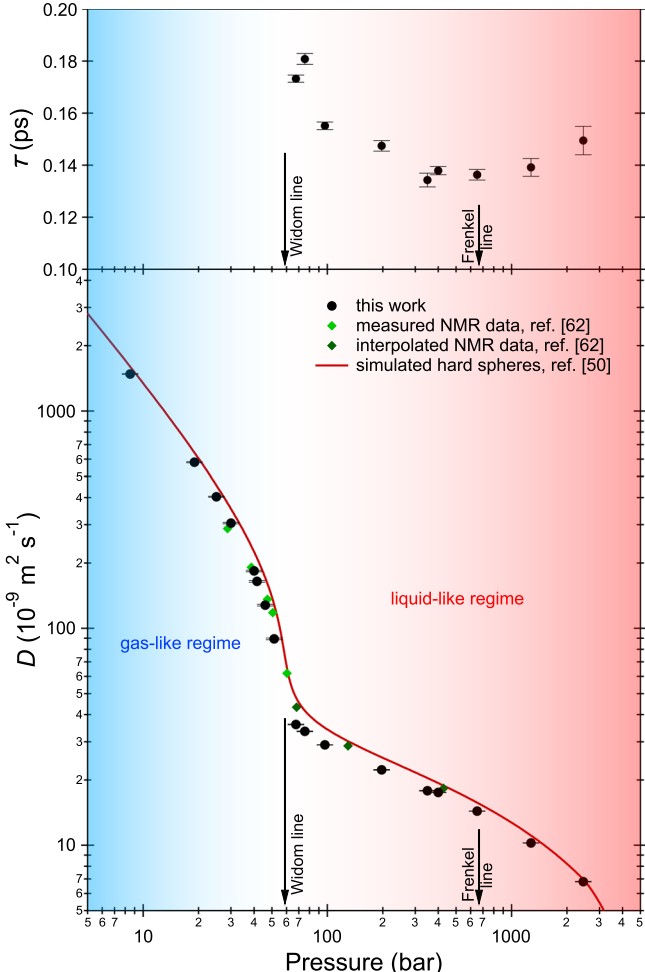

**Fig. 4 | Pressure dependence of the mean residence time and of the diffusion coefficient at 200 K.** Top panel: Parameter $\tau$ of the jump diffusion model as obtained from the fits shown in Fig. 3b. Bottom panel: Self-diffusion coefficient $D$ (full circles) as obtained from the fits shown in Fig. 3. Error bars correspond to one standard deviation, as obtained from the fits of ten or twelve data points shown in Fig. 3. For the diffusion coefficient at low pressures from the Gaussian fits, the error bars were obtained by propagating the errors in $K$ only. A gas-like (resp. liquid-like) regime is defined as being the regime where $D$ was obtained using Gaussian (resp. Lorentzian) fits and the two regimes are indicated with different colors, with the blur in between indicating that the Gaussian-to-Lorentzian transition is progressive. The crossover covers a pressure range from 40 to 80 bar at least. The arrows indicate the intercepts of the Widom and Frenkel lines with the 200 K isotherm. We also report i) experimental NMR values for methane at 200 K (full diamonds) measured in ref. 62 or obtained in ref. 62 from interpolation of data measured at other temperatures and ii) simulated values for the hard sphere fluid from ref. 50 (full line), plotted here using $T = 200$ K, $\sigma = 3.61$ Å, and the mass of the methane molecule.

at the same temperature of 200 K[62] and with a literature computational (molecular dynamics) prediction for hard spheres[50]. The hard-sphere prediction[50] reproduces the trend of the measured diffusion coefficient fairly well over the entire pressure range, and this for a $P$-independent value of the sphere diameter $\sigma$ of 3.61 Å. This is at variance with our earlier QENS work on methane[57], where we reached the much higher pressure of 1.4 GPa at 300 K and observed failure of the hard-sphere model.

The largest relative deviations between our data and the hard-sphere prediction are for the intermediate pressure range (40-80 bar), and this is likely to be associated to the existence of a pressure regime where both the Gaussian and Lorentzian modelling of the dynamic structure factor are not fully justified yet representing the only models, though approximate, presently available to describe the diffusive dynamics. Consistently, this is also the pressure range where the fits of the wavevector transfer dependence of the Gaussian and Lorentzian half widths are less satisfactory (see Fig. 3). However, while the Gaussian fits provide reasonable values for $D$, the Lorentzian fits provide unreasonably small values compared to the literature experimental values[62] and to the hard-sphere predictions[50] at all pressures below 67.5 bar. For example, at 51.5 bar, Lorentzian fits provide

$D = 50.7 \times 10^{-9} \text{m}^2\text{s}^{-1}$, to be compared with $89.4 \times 10^{-9} \text{m}^2\text{s}^{-1}$ for the Gaussian fits. This suggests that the Gaussian description is more appropriate than the Lorentzian one for all of our pressure points up to 51.5 bar. Above 51.5 bar, the situation is reversed, with i) the Gaussian fits providing unreasonably small diffusion coefficient values (for example at 67.5 bar, $36.0 \times 10^{-9} \text{m}^2\text{s}^{-1}$ for the Lorentzian and $23.5 \times 10^{-9} \text{m}^2\text{s}^{-1}$ for the Gaussian fits) and ii) the Lorentzian fits clearly describing the measured spectra much better (see Fig. 2). To further support our choice of employing Gaussian fits up to 51.5 bar and Lorentzian fits at and above 67.5 bar, various considerations including the likeliness of the obtained diffusion coefficient are listed and quantified in the Supplementary Information (Supplementary Note 1 and Figs. 3,

4), where we also report the results of a Bayesian analysis[63] of the experimental spectra (Supplementary Note 2 and Figs. 5–10).

The crossover we observe here is progressive and clearly not a first-order phase transition. It is conceivable that over a small pressure range that is enclosing the Widom line, the sample is heterogeneously made of variable fractions of gas-like and liquid-like molecules, with such fractions continuously varying with pressure/temperature, following a suggestion based on simulations[64]. However, a two-component model which is simply a sum of a Gaussian and a Lorentzian function would be most likely too simplified to correctly describe the present experiments, as it would correspond to neglecting molecular exchanges among gas-like and liquid-like clusters as well as the contribution of the interfaces to the measured dynamic structure factor. A recent theoretical study of water[65] indicated that molecular exchanges indeed occur among clusters of all scales along the Widom line. As for the contribution of the interfaces, they would be negligible only if the characteristic length scale of the spatial heterogeneity, which is not known for methane to our knowledge, was much larger than the (Å) length scale probed by our experiments. Still, this aspect of the data analysis certainly deserves further investigation in the future. Recently, neutron imaging experiments[17,18] succeed in observing the density fluctuations of supercritical water confined in a carbon porous monolith while the system evolves rapidly from liquid-like to gas-like as the Widom line is crossed during isobaric heating. The porous carbon monolith indeed delays water structural changes, and thus it brings the time scale of the liquid-like to gas-like transition at the accessible time scale of the experiment. The same authors also show that the transition from liquid-like to gas-like supercritical states when crossing the Widom line finds explanation based on the pseudo-boiling concept[17,18].

Finally, the self-diffusion coefficient can be employed to test semi-empirical relationships used in the literature, notably the Stokes–Einstein–Sutherland (SES) relation. We calculated the product of $D$ with the shear viscosity $\eta$ (see Supplementary Fig. 11), using the literature estimation of $\eta$ given in the NIST Chemistry WebBook, which uses ref. 66. The employed viscosity values are tabulated in Supplementary Table 1. The SES relation predicts a constant product $D\eta$ along isotherms. In our case, the product shows two regimes along the 200 K isotherm: below the pressure of intercept with the Widom line, $D\eta$ decreases with increasing pressure and the SES relation is violated, above that pressure, $D\eta$ is constant with pressure and the SES relation holds. A similar trend is obtained when using the experimental NMR values of $D$ from ref. 62 (see Supplementary Fig. 11). It must be said that the macroscopic viscosity values we used are those of a homogeneous methane fluid at the given pressure and temperature, but this approximation cannot be taken for granted close to the Widom line. In the literature, the SES relation is typically not expected to hold at low pressures (below twice the critical density). It is indeed a well-known result of isomorph theory[67,68], experiments of various simple fluids[69], and simulations of hard spheres and of the Lennard-Jones system[49,50] that the product $D\eta$ cannot be constant over isotherms in the gas phase. On the other hand, the SES relation commonly holds in the literature of atomic and molecular liquids, and $D\eta$ is indeed constant within the experimental uncertainties in compressed liquid methane at constant $T$, as shown for example in Supplementary Fig. 5 of our previous work[57] for literature experimental data at low temperatures[52]. Yet, $D\eta$ increases with pressure in very dense supercritical methane along the room-temperature isotherm[57].

## Discussion

The fundamental problem of distinguishing between a gas-like and a liquid-like region in the phase diagram of simple supercritical fluids has fed an intense debate in the literature[14–29,31–39,45,64,65,70–72]. In the present measurements, we establish the existence of a crossover in the self-dynamic structure factor of SCF methane measured by QENS along the 200 K isotherm. The change from a Gaussian to a Lorentzian lineshape in the measured signal reveals a crossover from a gas-like to a liquid-like diffusive dynamics of the molecules which occurs roughly at the pressure at which the Widom line intercepts the 200 K isotherm, indicating that the Widom line separates gas-like and liquid-like behaviours as concerns the self diffusion of $CH_4$ in the vicinity of the critical temperature. To our knowledge, the present work represents the first to employ the QENS technique to investigate a SCF sample in the vicinity of its critical point. The advantage of using neutrons to investigate the diffusion in a SCF resides in the fact that neutron scattering is a weak process and the sample response is linear. It follows that the measured dynamic structure factor is determined by the spectrum of the spontaneous correlations between molecules, and thus gives a fundamental information on the molecular interaction[73].

Along the thermodynamic path we followed, we could cross both the Widom and the Frenkel line. The picture that emerges is that at pressures below the Widom line, the molecular diffusion of methane is basically governed by the harsh repulsive forces between molecules while the weaker, longer-ranged attractive forces have little influence on the dynamics. The system can be described as a dense (non-ideal) gas. Conversely, above the Widom line, the sharp increase in density makes the free path of the molecules to strongly decrease and the diffusion mechanism becomes sensitive to the longer-ranged molecular interaction and depends on the average molecular distances. The molecular diffusion can be thus described as that of a liquid and the Stokes–Einstein–Sutherland relation holds. This happens starting from much lower densities (by approximately a factor of 2) than those typical for liquid methane. For the experimental determination of the self-diffusion coefficient as a function of the pressure, assumption of a hard-sphere behaviour was made and literature density values from the known equation of state of methane were used in addition to the Gaussian widths. The pressure dependence of the obtained diffusion coefficient shows a faster slowing down when crossing the Widom line, reminiscent of the faster increase in density. It can be remarked that molecular dynamics simulations[33] using the TIP2005 potential validated against experimental viscosity data somehow similarly showed a fast change in the molecular self-diffusion coefficient of supercritical water as a function of temperature at the crossover between the two (gas-like and liquid-like) regions. The curves remain continuous as expected in the supercritical region, but close to the critical point, they show a drastic change of slope at around the Widom line[33].

In our data analysis, we have used the only diffusive models available in the literature. We must highlight that over the crossover range, a more complex modelling for the dynamic structure factor is highly likely to be required. At present, we are not aware of an appropriate model to describe the behaviour in the intermediate pressure range marking the change between a gas-like to a liquid-like regime. In conclusion, the measured spectra are correctly reproduced by Gaussians or Lorentzians, except over an intermediate crossover range, which is the result of a situation that does not fit the assumptions for diffusion models that apply to perfect gases or liquids. A smooth transition is observed in between the two regimes, in the region where the Widom line intercepts the experimental isotherm. We cannot exclude the possibility that in this region, the system is actually heterogeneous and formed by gas-like and liquid-like clusters, as observed in supercritical water[17,18] and carbon dioxide[19]. However, to corroborate this hypothesis, enthalpy measurements or small-angle X-ray/neutron scattering experiments should be performed on a scale matching the correlation size of the clusters, which is unknown for methane. Furthermore, to our knowledge, no analytical model exists to model the self-dynamic structure factor of a bi-phasic system unless the aggregates of each phase have sufficiently large spatial extensions for the "volume" of the interfaces to be negligible, thus justifying the addition of two distinct and independent spectral shapes.

At considerably higher pressures, the presence of the Frenkel line does not seem to affect the pressure dependence of the diffusion coefficient nor the diffusion mechanism as probed in the present measurements, with the exception of a shallow minimum in the trend of the in-cage molecular residence time $\tau$ as a function of pressure. On the non-rigid fluid side of the Frenkel line, a competition between the repulsive potential and attractive Van der Waals forces between adjacent molecules undergoing diffusion is in place. Attractive Van der Waals forces dominate in the fluid at the lowest densities and lead the in-cage residence time of the molecules to decrease upon pressure increase[70]. Conversely, when crossing the Frenkel line, $\tau$ increases with pressure because of the major role of the repulsive potential and the consequent stabilisation of the neighbour's cage. As a consequence, the variation of the residence time as a function of the pressure in an isothermal experiment can display a minimum as one proceeds from the non-rigid to the rigid fluid. Future studies will be able to verify whether this intriguing observation is reproduced and matches the Frenkel line intercepts also along other isotherms. It was previously suggested that, at pressures above the Frenkel line, the (relatively) close packed arrangement of the molecules determines a much stronger interaction between neighbouring molecules as well as the onset of shear modes (see for example ref. [27]). Consequently, the diffusion mechanism progressively becomes more similar to the hopping observed in solids through vacancies and the system can be described as a rigid liquid on the short time scales such as those here explored[71].

Finally, concerning the Widom line, it is worth recalling that there is a group of lines emanating from the critical point which are associated with maxima/minima in different thermophysical properties such as the speed of sound, isothermal compressibility, isochoric and isobaric heat capacities, which have all been named "Widom lines"[72]. Close to the critical point, as in the present experiment, the choice of the specific thermophysical property is relatively unimportant as all of the maxima/minima closely coincide, but far away from the critical point, the maxima in the response functions begin to deviate significantly from each other. However, it is now generally agreed that the Widom line can be identified to a maximum in the correlation length characterizing the range of the molecular interactions which is here directly probed through the dynamic structure factor. In this extent, the dynamic structure factor turns out to be a pertinent physical observable allowing us to distinguish between a gas-like and a liquid-like behaviour within the supercritical fluid. Its change in lineshape when crossing the Widom line reflects a substantial change in the molecular interaction length scale and diffusive behaviour. This observation could be exploited to tailor the efficiency of supercritical fluids in industrial applications[74] and could have major consequences on the geodynamics of gas giant planets in our Solar System and beyond.

## Methods

### Experimental setup

The measurements have been carried out at the Institut Laue-Langevin large-scale facility in Grenoble, France, on the time-of-flight neutron spectrometer IN6-SHARP. The wavelength of the incoming neutron beam was set to 5.12 Å. Neutron scattering spectra of methane at $200 \pm 1$ K were recorded at pressures between 8.5 and 2450 bar with a typical acquisition time of 3–4 h per pressure point. The sample was contained in a cylindrical pressure cell of aluminium alloy whose height was about 50 mm and whose internal diameter was 6 mm, and was compressed using a gas compressor working with methane. In this setup, the sample volume does not change upon pressure increase/decrease and the number of methane molecules in the neutron beam increases with increasing pressure. To reduce the multiple scattering contributions, we inserted a cylindrical aluminium spacer of 5 mm in diameter for the measurements at the following eight pressures: 46,

97, 197, 349, 400, 650, 1275, and 2450 bar. The nine pressure points measured without aluminium spacer were: 8.5, 19, 25, 30, 40, 41.5, 51.5, 67.5, and 75.5 bar. The $CH_4$ (purity > 99.95%) bottle was purchased from AirLiquide, Saint Priest, France. Pressure was directly measured by a manometer attached to the capillary connecting the gas compressor with the high-pressure cell. The uncertainty on the pressure measurement was 1 bar. The sample temperature was controlled using a standard "orange" cryostat. Empty-cell spectra were measured with and without aluminium spacer.

### Data extraction and fitting

For each pressure point, constant-$Q$ spectra from 0.4 to 1.5 Å⁻¹ with 0.1 Å⁻¹ steps were extracted (except for Fig. 1 where a larger integration range of 0.3 Å⁻¹ was used). Their intensity was normalized using the measurement of a vanadium standard and the empty-cell signal was subtracted using $P$- and $Q$-dependent transmission values (the spectra reported in Figs. 1 and 2 are empty cell-subtracted).

The different possible motions of methane (vibrations, rotations and translations) lie on different time scales. The contribution of the vibrational motion is very fast and can be approximated with a Debye-Waller factor. Rotations are too fast to be properly resolved on IN6-SHARP and probably only contribute as a very broad quasi-elastic signal appearing as a flat background in our spectra. Therefore, only the vibrational and translational motions were included in the fitting model, similarly to our previous QENS study of dense supercritical methane at 300 K[57]. The accessible energy range is inherently limited in time-of-flight neutron spectrometers and is rather narrow at small $Q$. For example, it goes from −2.2 to 1.8 meV at 0.5 Å⁻¹, as can be seen in Fig. 2. To limit the effect that the rotational contribution might have on our results, fits of the spectra at high $Q$ were restricted over a reduced energy range.

Spectra are given in arbitrary units and were accordingly fitted using a Gaussian or Lorentzian function multiplied by a scaling factor in arbitrary units. A flat positive background and a zero shift (small difference between the values of the true and nominal zeros of the energy transfer axis) were included. Moreover, in the fitting of the spectra, convolution of $S_{\text{self}}(Q,\omega)$ with the instrumental resolution function was taken into account. The resolution function of the spectrometer $R(E)$ was obtained by fitting the measurement of the vanadium standard to a pseudo-Voigt distribution. Its full width at half maximum was 0.077 meV. Therefore, each constant-$Q$ spectrum was fitted using either of the two formulas:

$$I^{\text{Gauss}}(E) = I_0^{\text{Gauss}} \exp\left(-\frac{\ln(2)(E-E_0)^2}{\hbar^2 \Gamma_G^2}\right) \otimes R(E) + B_0, \tag{7}$$

$$I^{\text{Lor}}(E) = \frac{I_0^{\text{Lor}}}{\pi} \frac{\hbar \Gamma_L}{(E-E_0)^2 + \hbar^2 \Gamma_L^2} \otimes R(E) + B_0. \tag{8}$$

Both Gaussian and Lorentzian fits have four parameters: $I_0^{\text{Gauss}}$ or $I_0^{\text{Lor}}$, $\Gamma_G$ or $\Gamma_L$, $E_0$, and $B_0$. The zero shift of the energy-transfer axis $E_0$ was generally left free to vary. However, it was systematically imposed at high $Q$ in the Gaussian fits to an extrapolation of the small $Q$ values (it would otherwise converge to relatively large values up to 0.2–0.3 meV). Similarly, to ensure a smooth $Q$ dependence, the flat background $B_0$ was imposed to an extrapolation of the higher $Q$ values at small $Q$ values in the Gaussian fits and at the smallest $Q$ only in the Lorentzian fits (it would otherwise converge to larger values). We checked that both of these choices (for zero shift and flat background) have only minor effects on the determination of the diffusion coefficient. Supplementary Figs. 12 and 13 report all $B_0$ values. Intensities and widths were always left as free parameters in the fitting. Supplementary Fig. 14 reports the fitted Lorentzian half widths at pressures where Gaussian fits were preferred and the fitted Gaussian

half widths at pressures where Lorentzian fits were preferred (therefore complementing Fig. 3 of the main text).

The obtained intensities $I_0^{Gauss}$ and $I_0^{Lor}$ are plotted as a function of $Q$ in Supplementary Fig. 15. They were fitted using:

$$I_0^{Gauss} \propto \frac{1}{Q}\exp(-Q^2<u^2>/3), \tag{9}$$

$$I_0^{Lor} \propto \exp(-Q^2<u^2>/3). \tag{10}$$

$I_0^{Gauss}$ corresponds to the height of the Gaussian function before convolution with the instrumental resolution and is proportional to the factor $1/Q$ (see Eq. (3)) multiplied by the Debye-Waller factor $\exp(-Q^2<u^2>/3)$. $I_0^{Lor}$ corresponds to the area of the Lorentzian function before convolution with the instrumental resolution and is simply proportional to the Debye-Waller factor. The resulting values of $<u^2>^{1/2}$ are reported as a function of the pressure in Supplementary Fig. 16. They amount to roughly 1 Å, which is a reasonable value and within the range of values reported in the literature for example of liquid water at ambient conditions, for which many QENS studies exist. We observe an increasing trend for $<u^2>^{1/2}$ with increasing pressure (see Supplementary Fig. 16) in both estimations (from the Gaussian and Lorentzian intensities) with a discontinuity between the Gaussian and Lorentzian estimations whose origin is presently unclear.

The Gaussian half widths were fitted over the $Q$ range from 0.4 to 1.5 Å$^{-1}$, as shown in Fig. 3a. In the linear fits, the data points were not weighted according to the estimated error in $\Gamma_G(Q)$. This choice moderates the weight of the low-$Q$ points (having the smaller error), corresponding to the regime where the theoretical ground for the Gaussian modelling is weaker. The Lorentzian half widths were fitted over a slightly narrower $Q$ range from 0.4 to 1.3 Å$^{-1}$ (Fig. 3b), similarly to our previous work[57], in order to limit the potential effect of the rotational contribution. This is because the rotational contribution i) becomes slower with increasing pressure and might potentially approach the time scale window accessible by the present experiments at our highest pressures and ii) is known from analytical modelling to have increasing intensity with increasing $Q$. The data points were weighted according to the estimated error in $\Gamma_L(Q)$.

Finally, multiple scattering contributions to the spectra were ignored. The multiple scattering was calculated following the established procedure described in ref. 75 for a few selected pressure points and its subtraction was observed to have negligible effect on the fits and on the determination of the diffusion coefficient. This test was performed at three pressures: 67.5 bar (measured without the aluminium spacer in the high-pressure cell containing the sample), and 97 and 1275 bar (both measured with the spacer). The calculated multiple scattering is much weaker and broader than the single scattering and included in the flat background over the considered energy range. This is confirmed by the observation that there is no inconsistency between the results for the data measured with and without the aluminium spacer. In particular, the results for the point at 46 bar, which was measured with the spacer, perfectly interpolate in between the results for 41.5 and 51.5 bar, which were measured without the spacer.

## Data availability
All raw data were generated at the Institut Laue-Langevin large-scale facility and are publicly available from the ILL depository[76]. Derived data generated in this study are provided in the Supplementary Information. All other information can be obtained from the corresponding authors upon request. Source data are provided with this paper.

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

## Acknowledgements

We acknowledge the Institut Laue-Langevin for provision of beam time and Claude Payre, James Maurice, Eddy Lelièvre-Berna, Jean Marc Zanotti and Quentin Berrod for advice and technical assistance during the experiment. We are grateful to Helmut Schober for having authorised the experiment at the ILL reactor. We acknowledge Luisa Scaccia for her invaluable contribution to making available the Bayesian analysis code written with A.D.F. We also acknowledge Ubaldo Bafile, José Teixeira and Francesco Sciortino for useful discussions. U.R. thanks the UKRI for financial support through the Future Leaders Fellowship MR/V025724/1 held by D. Laniel. L.E.B. acknowledges funding through the Swiss National Fund (FNS) grant n 212889 and the ANR-23-CE30-0034 EXOTIC-ICE. For the purpose of open access, the authors have applied a Creative Commons Attribution (CC BY) licence to any Author Accepted Manuscript version arising from this submission.

## Author contributions

The project was conceived by F.F., F.A.G., M.S. and L.E.B. The experiments were performed by F.F., F.A.G., M.S., M.M.K. and L.E.B. New analytic tools have been provided by F.F. and L.E.B.; The data were analysed and the figures produced by U.R., with contributions from F.F. and L.E.B.; F.F. performed the Bayesian analysis using the code written by A.D.F. The manuscript was written by U.R. and L.E.B. All authors discussed the results and commented on the manuscript.

## Competing interests

The authors declare no competing interests.
