## [Peer Review File · Nature Communications]

Crossover from gas-like to liquid-like molecular diffusion in a simple supercritical fluidEditorial Note: This manuscript has been previously reviewed at another journal. This document only contains reviewer comments and rebuttal letters for versions considered at *Nature Communications*. Mentions of prior referee reports have been redacted.

REVIEWER COMMENTS

Reviewer #2 (Remarks to the Author):

The present manuscript provides a clear evidence that molecular diffusion in supercritical methane changes qualitatively from gas like to liquid like when crossing the Widom line, whereas nothing comparable happens upon crossing the Frenkel line. This work is a crucial puzzle piece to understand the molecular scale changes. This challenging work has been done with great care. The authors put significant effort to answer all the points raised by the reviewers. Most of them does not need further discussion. Nevertheless, I still recommend some further revision of the article. Presently, the sharp change in the diffusion coefficient is in the focus, although there is no available model to handle the cross-over region properly. Even if the cross-over range is narrow on the presented phase diagram, and can be indeed called as a sharp transition, it is misleading, since it is not a kind of first-order transition. Instead, literature and present data support a heterogeneous two component state close to the Widom line. Therefore, I suggest to emphasize the change itself, not it's sharpness, and discuss the cross-over RANGE together with present limitations of available theories and consequences e.g. on the calculated diffusion coefficient. I would also suggest to include the observation length scale in the discussion, which I feel to be quite important for the cross-over range. My detailed comments are below.

Answers to Reviewers:

[REDACTED]

The first two references and descriptions are exchanged.

[REDACTED]

I disagree here with the authors. First of all, the correlation length of the spatial heterogeneity should be considered, which is not known to my knowledge. Maxim et al. observes separation on the mm scale (although using a porous confining medium, therefore not necessarily valid here), whereas Pipich et al. on a smaller scale, but for CO₂, and not for water. The correlations probed by Sun et al. and in this manuscript are on an even smaller scale. Heterogeneity beyond the probed scale contributes to the spectra in an additive manner. Therefore the decomposition into two-components by Sun et al. is to my opinion meaningful and clearly supported by their measurement. The negligibility of interfaces depends on the correlation length, which is unknown. In general, the interface is likely to behave differently both from the liquid- and gas-like components, and if the probed length scale is shorter than the correlation length (i.e. the two component model is a reasonable description for the bulky interior of the domains), the interface could give rise to a smeared spectrum. Actually, the existence of interface could be the reason, why Sun et al. could not fit the line-shapes with usual models, but had to do a phenomenological fit instead. Their data is clearly against a smooth, homogeneous transition. Actually, this two

component model is what I suggested in my previous review as alternative fit model. I do not see it as essential for different reasons, and I agree that it is also too idealistic because of the interface, however, the relative intensities could be interesting for future studies to compare with. On the other hand, the article by Sun et al. does not reduce the relevance of this manuscript. Contrary, they are highly complementary, since they do not probe diffusion, "only" vibrations, i. e. shorter time scales! Obviously, in a heterogeneous medium the diffusion will depend on the probed length (and time) scale. Since QENS probes the same length scale as Sun et al., it is expected that water diffusion in the corresponding range of the phase diagram should be described by a two component model. Whereas diffusion over a length scale bigger than the correlation length, the data would have to be described by one component only, where diffusion is timewise alternating between gas-like and liquid-like behaviour. I'm not aware of a corresponding analytical model. The situation is even more complicated if the probed and correlation lengths are comparable. A complete picture could be obtained for water if correlation length and diffusion would be also measured. The present manuscript investigates diffusion in methanol, for which the correlation length is not known, and likely to be different from water. Therefore, even if the same length scale is probed, as by Sun et al., it can't be predicted if the two component model would be more suitable or not.

And a related answer to one of my questions:

[Redacted]

Reading this, we probably have more similar thoughts than expressed by words?

[Redacted]

See above, for transport, e.g. self-diffusion this seems to be indeed the first article, but not for dynamics in general.

[Redacted]

There is actually a very recent article, which could be cited here:

A. Tahaei et al: Scaling Description of Dynamical Heterogeneity and Avalanches of Relaxation in Glass-Forming Liquids, PHYSICAL REVIEW X 13, 031034 (2023), Scaling Description of Dynamical Heterogeneity and Avalanches of Relaxation in Glass-Forming Liquids

Maybe even some further theoretical descriptions can be "borrowed" from glass physics?

I would like that the authors state clearly in the manuscript that the change in the measured spectrum is actually small, and that the significant change in D comes from the different calculation, basically from the change in density. QENS probes diffusion on the atomic scale, where the system is heterogeneous, therefore I do not find it proper to use a density value, which is only a macroscopic average. I understand that there is no better choice available, but this rough approximation has to be stated. The same applies to the viscosity, see below. Please also put a figure into the SI showing the used density and viscosity as a function of pressure, for full transparency.

On this small length scale, where the system is most likely heterogeneous, it is only a crude approximation to take a macroscopic average for viscosity.

Given these uncertainties, and without additional information about the heterogeneity and without a proper model, I see it as follows:

The improved manuscript provide now a solid evidence for the change in line shape (and therefore in the self-diffusion), i.e. it shows that the cross over from gas-like to liquid-like behavior happens at the Widom line. This should be the central point of the manuscript. However, we should not forget that QENS measures diffusion on a local scale, therefore it is not evidenced that macroscopic, long range transport shows also a cross-over there. The further discussion of D and break-down of Stokes-Einstein relation is based on the crude approximation using macroscopic (or average) density and viscosity, although the system seems to be heterogeneous at the probed length scale, at least in the vicinity of the Widom line. The lack of suitable theory does not mean that the derived conclusions using those approximations are correct. Therefore, that part of the manuscript should not be taken as evidence but only as a discussion to explore possible consequences and provide a motivation to develop suitable theories.

Manuscript:

- Abstract: "Concurrently, a sharp change in the pressure dependence of the molecular self-diffusion coefficient takes place." and Intro "The molecular diffusion coefficient has been determined over a range of densities covering almost three orders of magnitude and compared to both (non-ideal) gas and dense-fluid theoretical descriptions. Its variation as a function of pressure

shows a clearcut slowing down when crossing the Widom line, reminiscent of the change in the fluid compressibility.”

I see it necessary to relativate these statements. It is clearly shown that far enough from the Widom line the system is a gas and a fluid, respectively. However a “sharp change” or “clearcut” at the Widom line is only seen, when the one or the other state is enforced in the modeling, although the real case seems to be a heterogeneous two component mixture (where theoretical models are missing), and consequently probably a smoother transition. Contrary, the statement above suggest an abrupt change, which is not what has been observed.

“This result is exclusively based upon the experimental determination of a basic transport property as a function of the pressure and use the only available diffusive models available in the literature, without resorting to any other external input or calculations.”

I suggest to rephrase this sentence. QENS measures diffusion on the molecular scale, which is not necessarily a transport property. In the liquid and in the gas phase yes, but not without further theory in the heterogeneous two component state. Something similar to “This result is exclusively based upon the experimental determination of a basic transport property as a function of the pressure. Although, the self-diffusion coefficient changes in reality probably smoother than presented, but we are not aware of an appropriate model to describe the likely heterogeneous behavior in that range.”

- Fig. 4: the cross-over range should be clearly indicated for the effected data points, not only as shaded background, since their value has a corresponding much higher (unknown) error.

Reviewer #3 (Remarks to the Author):

In response to the reply to my previous report:

- The authors:

[REDACTED]

Based on:

[REDACTED]

These are two different issues. I reckon the debate Widom vs Frenkel demarcation line. But what I said is that the existence of two regions is nowadays well established in literature, including most of the papers in the list provided by the authors. And it is such existence and its implications that the authors were selling in the first and last sentence of the abstract. This motivated my comment.

- Forcing the peak position and the background to be the same for the two lineshapes would be a mistake, as it would make both Gaussian and Lorentzian fits less satisfactory and thus the estimation of the diffusion coefficient less precise.

The key point of the whole discussion is to ascertain which of the two lineshapes does a better job on the same spectrum. Constraining parameters directly related to quantitative spectral features such as background and linewidth would just make the comparison more stringent. Adding free parameters obviously improves the agreement with the data, but again what is needed here is a relative comparison between two lineshapes. And such comparison makes sense when they share the same key parameters describing well defined spectral features. What is the value of a comparison between two FWHM coming from lineshapes with different background, when one of the two is clearly unrealistic (as in Fig.1 @97bar) ? Again, the real problem here is in the inherent limitation of QENS in terms of spectral coverage, acknowledged by the authors themselves.

Finally, the authors make a strong point on the value added by the experimental nature of their

work, as most of the work in favor of the Widom line is theoretical and numerical while the present manuscript would be only the fourth experimental paper supporting the Widom line. Very honestly, I consider this argument very weakening. I still have the same initial concerns on the reliability of this work's conclusions, mostly for the same reasons expressed by referee 1 and in part by referee 2. But even assuming that everything is solid and robust, why "experimental Methane" should be the smoking gun solving the entire puzzle? What is wrong with other systems obeying model potentials? The general argument about a presumed superiority of experimental data with respect to numerical studies sounds like an inappropriate bias in this specific context looking for a universal behavior.

In my previous report, when the MS was under consideration in Nature Physics, I recommended publication in a more specialized journal, after strengthening and clarifying the data analysis. I suggested a possible strategy. The authors resubmitted to Nature Communications, a multidisciplinary journal, and claim that my indication for a possible improvement of the data analysis is a wrong one.

I reaffirm here my opinion on this work. Hence, I cannot recommend publication of the manuscript in its present form.

We wish to thank both reviewers for providing useful suggestions to improve the quality of our work.

Below, reviewers' comments are shown in blue and our responses are in black (and in a different font type).

In the attached version of the main text of the manuscript, all changes addressing the concerns of the reviewers (or editorial requests) are shown in red. Other minor changes are shown in green.

Reviewer #2 (Remarks to the Author):

The present manuscript provides a clear evidence that molecular diffusion in supercritical methane changes qualitatively from gas like to liquid like when crossing the Widom line, whereas nothing comparable happens upon crossing the Frenkel line. This work is a crucial puzzle piece to understand the molecular scale changes. This challenging work has been done with great care. The authors put significant effort to answer all the points raised by the reviewers. Most of them does not need further discussion. Nevertheless, I still recommend some further revision of the article. Presently, the sharp change in the diffusion coefficient is in the focus, although there is no available model to handle the cross-over region properly. Even if the cross-over range is narrow on the presented phase diagram, and can be indeed called as a sharp transition, it is misleading, since it is not a kind of first-order transition. Instead, literature and present data support a heterogeneous two component state close to the Widom line. Therefore, I suggest to emphasize the change itself, not it's sharpness, and discuss the cross-over RANGE together with present limitations of available theories and consequences e.g. on the calculated diffusion coefficient. I would also suggest to include the observation length scale in the discussion, which I feel to be quite important for the cross-over range. My detailed comments are below.

We thank reviewer #2 for appreciating our work and recommending publication after further revision, and for the very useful comments and suggestions.

Answers to Reviewers:

[REDACTED]

The first two references and descriptions are exchanged.

We apologize for this exchange in our previous reply. Both references are cited correctly in the manuscript.

[REDACTED]

[REDACTED]

I disagree here with the authors. First of all, the correlation length of the spatial heterogeneity should be considered, which is not known to my knowledge. Maxim et al. observes separation on the mm scale (although using a porous confining medium, therefore not necessarily valid here), whereas Pipich et al. on a smaller scale, but for CO₂, and not for water. The correlations probed by Sun et al. and in this manuscript are on an even smaller scale. Heterogeneity beyond the probed scale contributes to the spectra in an additive manner. Therefore the decomposition into two-components by Sun et al. is to my opinion meaningful and clearly supported by their measurement. The negligibility of interfaces depends on the correlation length, which is unknown. In general, the interface is likely to behave differently both from the liquid- and gas-like components, and if the probed length scale is shorter than the correlation length (i.e. the two component model is a reasonable description for the bulky interior of the domains), the interface could give rise to a smeared spectrum. Actually, the existence of interface could be the reason, why Sun et al. could not fit the line-shapes with usual models, but had to do a phenomenological fit instead. Their data is clearly against a smooth, homogeneous transition. Actually, this two component model is what I suggested in my previous review as alternative fit model. I do not see it as essential for different reasons, and I agree that it is also too idealistic because of the interface, however, the relative intensities could be interesting for future studies to compare width.

On the other hand, the article by Sun et al. does not reduce the relevance of this manuscript. Contrary, they are highly complementary, since they do not probe diffusion, “only” vibrations, i. e. shorter time scales! Obviously, in a heterogeneous medium the diffusion will depend on the probed length (and time) scale. Since QENS probes the same length scale as Sun et al., it is expected that water diffusion in the corresponding range of the phase diagram should be described by a two component model. Whereas diffusion over a length scale bigger than the correlation length, the data would have to be described by one component only, where diffusion is timewise alternating between gas-like and liquid-like behaviour. I’m not aware of a corresponding analytical model. The situation is even more complicated if the probed and correlation lengths are comparable. A complete picture could be obtained for water if correlation length and diffusion would be also measured. The present manuscript investigates diffusion in methane, for which the correlation length is not known, and likely to be different from water. Therefore, even if the same length scale is probed, as by Sun et al., it can’t be predicted if the two component model would be more suitable or not.

In the current version of the manuscript, we describe the relevant literature (page 4) as well as the possible spatial heterogeneity of our sample on the probed length scale and close to the Widom line as a possible limitation of our data analysis (pages 8-9 and 11). Moreover, the approximation related to using macroscopic density and viscosity values is mentioned at pages 7 and 9, respectively (see also another answer of ours below).

And a related answer to one of my questions:

[REDACTED]

[REDACTED]

Reading this, we probably have more similar thoughts than expressed by words?

We do believe that we have a similar opinion on this point, which was probably not clearly emerging from our manuscript. We hope that in the new version of the manuscript, this point has been made clearer.

[REDACTED]

See above, for transport, e.g. self-diffusion this seems to be indeed the first article, but not for dynamics in general.

We agree. We found there was one use of the expression ‘dynamical crossover’ in the manuscript and we modified it. We now write everywhere in the manuscript that this is the first article for “self-diffusion” or “self dynamics”, as requested by the reviewer.

[REDACTED]

There is actually a very recent article, which could be cited here:
A. Tahaei et al: Scaling Description of Dynamical Heterogeneity and Avalanches of Relaxation in Glass-Forming Liquids, PHYSICAL REVIEW X 13, 031034 (2023), Scaling Description of Dynamical Heterogeneity and Avalanches of Relaxation in Glass-Forming Liquids

Maybe even some further theoretical descriptions can be “borrowed” from glass physics?

We think that the dynamics of glass-forming liquids is a very different scenario from the one described in our manuscript. Using average macroscopic viscosity values, breakdown of the Stokes-Einstein-

Sutherland relation is commonly observed in glass-forming liquids when approaching the glass transition, as a consequence of the settling of dynamical heterogeneities, whose amplitude increase as viscosity grows, reaching very large values close to the dynamical arrest. In our case, we probe a rather low-viscosity regime and limited viscosity variation compared to a glass-forming liquid. Furthermore, in the present case, the two different pressure regimes for the product $D*\eta$ are observed far away from the Widom line, while the spatial and possibly dynamical heterogeneities would only involve a small pressure range close to the Widom line.

We must also reiterate an argument from our previous reply: Supplementary Figure 11 is only a minor result of this manuscript and the observed trend with pressure was expected. The very simple hard sphere model is enough to reproduce the trend (see Fig. 12 of Pieprzyk et al. DOI: 10.1039/c9cp00903e). We are afraid that using theoretical descriptions from glass physics would fall beyond the scope of the present work. This is possibly a valid suggestion for future theoretical investigations.

[REDACTED]

I would like that the authors state clearly in the manuscript that the change in the measured spectrum is actually small, and that the significant change in D comes from the different calculation, basically from the change in density.

We have now stated clearly in the manuscript that the change in the measured spectrum is small (page 5). We also state clearly (more than once, including in the abstract) that literature density values are needed for the determination of D in addition to the Gaussian widths.

QENS probes diffusion on the atomic scale, where the system is heterogeneous, therefore I do not find it proper to use a density value, which is only a macroscopic average. I understand that there is no better choice available, but this rough approximation has to be stated. The same applies to the viscosity, see below. Please also put a figure into the SI showing the used density and viscosity as a function of pressure, for full transparency.

We now stated this approximation for both the density and the viscosity (pages 7 and 9).

For full transparency, we have made a table reporting the used density and viscosity values (Supplementary Table 1).

On this small length scale, where the system is most likely heterogeneous, it is only a crude approximation to take a macroscopic average for viscosity.

Given these uncertainties, and without additional information about the heterogeneity and without a proper model, I see it as follows:

The improved manuscript provide now a solid evidence for the change in line shape (and therefore in the self-diffusion), i.e. it shows that the cross over from gas-like to liquid-like behavior happens at the Widom line. This should be the central point of the manuscript. However, we should not forget that QENS measures diffusion on a local scale, therefore it is not evidenced that macroscopic, long range transport shows also a cross-over there. The further discussion of D and break-down of Stokes-Einstein relation is based on the crude approximation using macroscopic (or average) density and viscosity, although the system seems to be heterogeneous at the probed length scale, at least in the vicinity of the Widom line. The lack of suitable theory does not mean that the derived conclusions using those approximations are correct. Therefore, that part of the manuscript should not be taken as evidence but only as a discussion to explore possible consequences and provide a motivation to develop suitable theories.

We entirely agree with the comment of the referee. The central point of the manuscript is the change in lineshape, which shows that the crossover from gas-like to liquid-like behaviour happens at the Widom line.

The Gaussian-to-Lorentzian change is in the raw data themselves and is independent on the details of the data analysis. In the broad experimental literature (and following established textbook knowledge), QENS data are fitted using a Gaussian function for gas samples and using a Lorentzian function for liquid samples.

The sharp change in the diffusion coefficient with pressure (mentioned in the abstract in the previous version of the manuscript) is not in the focus anymore, see our next answer just below.

Manuscript:

- Abstract: "Concurrently, a sharp change in the pressure dependence of the molecular self-diffusion coefficient takes place." and Intro "The molecular diffusion coefficient has been determined over a range of densities covering almost three orders of magnitude and compared to both (non-ideal) gas and dense-fluid theoretical descriptions. Its variation as a function of pressure shows a clearcut slowing down when crossing the Widom line, reminiscent of the change in the fluid compressibility."

I see it necessary to relativate these statements. It is clearly shown that far enough from the Widom line the system is a gas and a fluid, respectively. However a “sharp change” or “clearcut” at the Widom line is only seen, when the one or the other state is enforced in the modeling, although the real case seems to be a heterogeneous two component mixture (where theoretical models are missing), and consequently probably a smoother transition. Contrary, the statement above suggest an abrupt change, which is not what has been observed.

We have made the requested changes. However, we want to mention that a fast change in the pressure dependence of the self-diffusion coefficient is also visible in the 50-years old and very precise literature (NMR) data of methane (ref. 62 of the manuscript) at 200 K, as reported in our Fig. 4, as well as at 195 K from the same NMR reference. Also, the NMR diffusion coefficient values at 195 K from ref. 62 are almost perfectly reproduced by the recent computational (MD) study by Singer et al [DOI: 10.1063/1.5027097].

In the abstract, “clear” was removed from “we observe a clear crossover in the measured dynamic structure factor from a gas-like Gaussian to a liquid-like Lorentzian signal”. Also, the entire sentence “a sharp change in the pressure dependence of the molecular self-diffusion coefficient takes place” was removed.

In the introduction, “clear” was modified into “gradual” in the sentence “[...] clear change from a Gaussian lineshape, describing the molecular diffusion in a gas, to a Lorentzian lineshape, describing the molecular diffusion in a liquid”.

Also, the sentence “its variation as a function of pressure shows a clearcut slowing down when crossing the Widom line, reminiscent of the change in the fluid compressibility” has been modified into “The pressure dependence of the obtained diffusion coefficient shows a faster slowing down when crossing the Widom line, reminiscent of the faster increase in density.” It has also been moved from the Introduction to the ‘Final comments’ section.

See also our answer just below for more on the “abrupt change in D ” issue.

“This result is exclusively based upon the experimental determination of a basic transport property as a function of the pressure and use the only available diffusive models available in the literature, without resorting to any other external input or calculations.”

I suggest to rephrase this sentence. QENS measures diffusion on the molecular scale, which is not necessarily a transport property. In the liquid and in the gas phase yes, but not without further theory in the heterogeneous two component state. Something similar to “This result is exclusively based upon the experimental determination of a basic transport property as a function of the pressure. Although, the self-diffusion coefficient changes in reality probably smoother than presented, but we are not aware of an appropriate model to describe the likely heterogeneous behavior in that range.”

We agree about the lack of a model to properly describe the pressure range of the crossover and we now discuss this extensively at page 10-11 of the manuscript. This was already briefly mentioned in the manuscript since its first version. However, we disagree with the reviewer’s suggestion that “the self-diffusion coefficient probably changes smoother than presented”, for the following reasons: i) Both the Gaussian and Lorentzian models seem to underestimate the diffusion coefficient over the pressure range of the crossover (when comparing our D values with the literature NMR data). It is not like if one model was overestimating D and the other one was underestimating it. ii) The fast slowing down of D at around the Widom line intercept is the expected consequence of the fast change in density (as we now mention

in the ‘Final comments’ of the manuscript). Below the critical pressure, an almost constant product between D and the density is a well-known result of the literature of all fluids, particularly at low temperatures (see for example DOI: 10.1021/acs.jced.5b00323). iii) The fast slowing down of D is reproduced by the literature simulations for hard spheres reported in Fig. 4. iv) Finally, switching from plotting D vs pressure (as in Fig. 4) to plotting D vs density makes the fast change disappear, as shown below:

Yet, we must admit that highlighting the sharpness in the pressure dependence of the diffusion coefficient at the Widom line intercept as in the previous version of the manuscript could have been misleading for the reader. To fix this, in the current revision of the manuscript, such a mention of the pressure dependence of the self-diffusion coefficient has been removed from the abstract and from the Introduction, and only appears in the ‘Final comments’ section (where it is accompanied by a warning on the lack of an appropriate model at the intermediate pressures). Again, as we have agreed with the reviewer a few answers above, this is not the central point of the manuscript.

The reviewer also comments on the use of “transport property” so in the ‘Final comments’, the sentence “Its change in lineshape when crossing the Widom line reflects a substantial change in the molecular interaction length scale with clear consequences in the transport properties of the system” has been modified.

- Fig. 4: the cross-over range should be clearly indicated for the effected data points, not only as shaded background, since their value has a corresponding much higher (unknown) error.

We have added a clarification to the caption of Fig. 4.

Reviewer #3 (Remarks to the Author):

In response to the reply to my previous report:

- The authors:

[REDACTED]

Based on:

[REDACTED]

These are two different issues. I reckon the debate Widom vs Frenkel demarcation line. But what I said is that the existence of two regions is nowadays well established in literature, including most of the papers in the list provided by the authors. And it is such existence and its implications that the authors were selling in the first and last sentence of the abstract. This motivated my comment.

We must say we are quite happy to read this comment by reviewer #3! Apparently, we misunderstood the reviewer on this matter during the first round of peer review.

The abstract was meant to provide the general reader, say a reader from another scientific field, with the bigger scientific picture. We tried to comply with the official guidelines of the journal (mentioning that the abstract should be only 150 words long) and had not much space for details. Of course, we did not intend to claim primacy over the existence of two regions in the supercritical state! This must be hopefully very clear from the rest of the manuscript.

In any case, the two sentences of the abstract have now been clarified. The new abstract is 250 words long and we hope this is ok with the Editor.

- Forcing the peak position and the background to be the same for the two lineshapes would be a mistake, as it would make both Gaussian and Lorentzian fits less satisfactory and thus the estimation of the diffusion coefficient less precise.

The key point of the whole discussion is to ascertain which of the two lineshapes does a better job on the same spectrum. Constraining parameters directly related to quantitative spectral features such as background and linewidth would just make the comparison more stringent. Adding free parameters obviously improves the agreement with the data, but again what is needed here is a relative comparison between two lineshapes. And such comparison makes sense when they share the same key parameters describing well defined spectral features. What is the value of a comparison between two FWHM coming from lineshapes with different background, when one of the two is clearly unrealistic (as in Fig.1 @97bar) ? Again, the real problem here is in the inherent limitation of QENS in terms of spectral coverage, acknowledged by the authors themselves.

We reiterate our argument from our previous reply: Ascertaining which of the two lineshapes does a better job on the same spectrum is only a delicate task at the intermediate pressures, very close to the Widom line intercept. We carefully address this issue in Supplementary Note 1.

There is no doubt that the Gaussian-to-Lorentzian crossover is in the pressure range of the Widom line. Even though the change is progressive and covers a pressure range from about 40 bar to 80 bar, the Frenkel line is very significantly far away (670 bar)! Moreover, in the intermediate region, we added a Bayesian analysis to get rid from any arbitrariness in the choice of the right function to use.

In addition to the measured Gaussian or Lorentzian lineshape of the energy spectra, the Q dependence of the fitted widths is also very different comparing the low and the high pressures: linear at low pressures (as predicted by the textbook ideal-gas model in Eq. 1) and close-to-quadratic at high pressures (as predicted by the textbook jump diffusion model for liquids in Eq. 6).

Finally, the authors make a strong point on the value added by the experimental nature of their work, as most of the work in favor of the Widom line is theoretical and numerical while the present manuscript would be only the fourth experimental paper supporting the Widom line. Very honestly, I consider this argument very weakening. I still have the same initial concerns on the reliability of this work's conclusions, mostly for the same reasons expressed by referee 1 and in part by referee 2. But even assuming that everything is solid and robust, why "experimental Methane" should be the smoking gun solving the entire puzzle? What is wrong with other systems obeying model potentials? The general argument about a presumed superiority of experimental data with respect to numerical studies sounds like an inappropriate bias in this specific context looking for a universal behavior.

Our manuscript never claimed superiority of experimental data with respect to numerical studies, but we do believe that experimental observation of a theoretical prediction is due in order to corroborate or falsify the prediction! For example, the study of Pipich *et al.* shows a comparison of the Frenkel lines in Fig. 1 of their paper underlining as MD simulations can deliver very uncertain results, which seem to strongly depend on the potential/criterion used to define the gas-like to liquid-like crossover. This situation particularly underlines the relevance of experimental proof such as performed in the present paper. But we admit we could have been more moderate in some of our statements of our previous reply to the referees. Our study is not solving the *entire* puzzle and more works will follow after this one.

In the revised version of the manuscript, we have worked on the text in order to present the results in a more balanced way. The insights that have been obtained by previous, including theoretical, work have been better acknowledged. It should be now clear in the new version of the manuscript that we are not claiming our study is solving the entire puzzle.

We think that the first observation of a gas-like to liquid-like crossover in the self dynamics of a supercritical fluid at the Widom line intercept is certainly worth publication in a high-impact journal. While a large number of theoretical studies exist in the literature on the Widom and Frenkel lines, we are not aware of any theoretical prediction of a Gaussian-to-Lorentzian change in the dynamic structure factor when crossing the Widom line, and we believe that our study can stimulate further theoretical work un such direction.

Given that reviewer #3 now acknowledges the literature "debate Widom vs Frenkel demarcation line" and that reviewer #2 states i) that "this is the first article for 'self-diffusion' or 'self dynamics'" and ii) that the article "shows that the cross over from gas-like to liquid-like behavior happens at the Widom line", why not giving this manuscript the visibility offered by publication in Nature Communications?

In my previous report, when the MS was under consideration in Nature Physics, I recommended publication in a more specialized journal, after strengthening and clarifying the data analysis. I suggested a possible strategy. The authors resubmitted to Nature Communications, a multidisciplinary journal, and claim that my indication for a possible improvement of the data analysis is a wrong one.

I reaffirm here my opinion on this work. Hence, I cannot recommend publication of the manuscript in its present form.

We appreciate the feedback provided by all reviewers and we take it very seriously. Many changes have been made to the manuscript after the first round of peer review in response to the criticism by all reviewers. This included a Bayesian analysis of the data which required significant amount of work and time. More changes have also been made in the current revision.

Reviewers' comments:

Reviewer #2 (Remarks to the Author):

The authors addressed all comments very carefully and implemented the requests satisfactorily. The manuscript highlights now the key findings well and has a well-balanced discussion. I'm happy with the outcome and recommend the publication in its present form.

Reviewer #3 (Remarks to the Author):

I still disagree on a crucial point of the methodology (the fitting strategy). The authors themselves reckon in the opening of their supplementary note 1 that "The Gaussian-to-Lorentzian crossover cannot be easily located in pressure based on the comparison of the chi-square values of the fits of the spectra alone". What else one should look at upon determining the best fitting lineshapes out of two candidates? So they try to make a strong point on the diffusion coefficient: "...being Q-independent, the likeliness of the diffusion coefficients obtained from the Gaussian and Lorentzian fits is a rather strong indication of the crossover."

But the diffusion coefficient is derived by the fitting parameters! And most important, the authors arbitrarily decide to extract D from the gaussian fit below 60 bar and from the Lorentian above 60 bar. Then it is obviously not surprising to find a crossover around 60 bar as result of this bias! Hence my initial suggestion of constraining the two lineshapes to the same critical parameters, positions, linewidths and background.

Concerning the value of experimental data vs simulations, in this specific contest: that one MD paper delivers very uncertain results means to me that, in absence of technical flaws, for that model system the crossover nature is unclear. Exactly as it can happen for any experimental result on a specific system, like the one reported in this manuscript.

I reckon that the authors have softened some key statements and conclusiveness of their manuscript. This is good and points to the direction I indicated in my previous report. The manuscript could be published in a specialized journal, in such case the audience itself will have the instruments to judge the technical points leading to the claims, as well as the relevance of such claims within the context of the vast literature dealing with this topic.

Below, Reviewers' comments are shown in blue and our responses are in black (and in a different font type).

Reviewer #2 (Remarks to the Author):

The authors addressed all comments very carefully and implemented the requests satisfactorily. The manuscript highlights now the key findings well and has a well-balanced discussion. I'm happy with the outcome and recommend the publication in its present form.

We would like to thank Reviewer #2 for recommending our manuscript for publication.

Reviewer #3 (Remarks to the Author):

I still disagree on a crucial point of the methodology (the fitting strategy). The authors themselves reckon in the opening of their supplementary note 1 that "The Gaussian-to-Lorentzian crossover cannot be easily located in pressure based on the comparison of the chi-square values of the fits of the spectra alone". What else one should look at upon determining the best fitting lineshapes out of two candidates? So they try to make a strong point on the diffusion coefficient: "...being Q-independent, the likeliness of the diffusion coefficients obtained from the Gaussian and Lorentzian fits is a rather strong indication of the crossover."

But the diffusion coefficient is derived by the fitting parameters! And most important, the authors arbitrarily decide to extract D from the gaussian fit below 60 bar and from the Lorentian above 60 bar. Then it is obviously not surprising to find a crossover around 60 bar as result of this bias! Hence my initial suggestion of constraining the two lineshapes to the same critical parameters, positions, linewidths and background.

We appreciate the Reviewer's thorough review of our manuscript. However, upon careful consideration of her/his comments, we believe there may have been a misunderstanding or oversight regarding certain points addressed in the manuscript (particularly, Supplementary Notes 1 and 2). Below, we have endeavored to clarify these aspects and provide further elucidation on our fitting strategy, for example in order to present more compelling arguments regarding our unbiased determination of the diffusion coefficient D parameter from the experimental data. The Reviewer is right when she/he observes that "the diffusion coefficient is derived by the fitting parameters", however what she/he seems to overlook is that the choice of the fitting model is far from being arbitrary; rather, it is grounded in robust statistical indicators. The statement "the authors arbitrarily decide to extract D from the gaussian fit below 60 bar and from the Lorentian above 60 bar" is wrong.

We used a Bayesian analysis (detailed in Supplementary Note 2) to discern the predominant spectral character – whether Gaussian or Lorentzian – for our experimental spectra. It is important to acknowledge that neither the Gaussian nor Lorentzian lineshapes accurately describe the state of the supercritical fluid within the intermediate pressure range close to the Widom line intercept, a fact extensively discussed in the manuscript and in our previous replies to Reviewer #2, who concurred that no alternative analytical description is presently available for application within this transitional range.

We wish to emphasize that the precise determination of the pressure threshold at which the crossover from a Gaussian-like to a Lorentzian-like line occurs cannot be solely discerned based on the chi-square values of the fits ONLY within the limited range of 46 – 67.5 bar. This range encompasses three experimental points at pressures of 46, 51.5, and 67.5 bar. On the other hand, it is unequivocally evident, as depicted in Figs. 1 and 2, and discussed in the text of the manuscript, that the superiority of either Gaussian or Lorentzian fits is distinctly discernible at pressures below 40 bar and above 67.5 bar, respectively. We would like to summarize here our main evidences for the presence and location of the Gaussian-to-Lorentzian crossover in our data.

- First, we observe, as illustrated in Supplementary Fig. 1 and discussed in Supplementary Note 1, a clear crossing in energy of the spectral lines at constant Q within the pressure range of 46 – 67.5 bar, indicating an unequivocal change in spectral shape independent of any fitting model or procedure. If the spectral line remained the same, no crossing would be observed with changing pressure.

- Second, the application of a Lorentzian fit to the data at all pressures below 60 bar yields a physically untenable negative background parameter (as illustrated in Supplementary Fig. 13 for 46 bar and 51.5 bar), leading to the dismissal of this fitting choice. The application of a Gaussian fit to the data at all pressures above 60 bar yields a too large background parameter (as shown for example in Fig. 1 for 97 bar), also leading to the dismissal of this fitting choice.

- Third, as can be seen in Fig. 3, the low- and high-pressure regimes are not only different in the lineshape of the measured spectra but also in the Q dependence of the fitted widths of the signal: at low pressure, Gaussian widths are linear in Q (as predicted by the textbook ideal-gas model in Eq. 1 of the main text) and at high pressure, Lorentzian widths are close-to-quadratic in Q (as predicted by the textbook jump diffusion model for liquids in Eq. 6 of the main text).

- Fourth, the diffusion coefficient parameter obtained from Lorentzian fits at all pressures below 60 bar deviates very significantly from the available NMR literature data (see page 8 of the main text), while the parameter obtained from Gaussian fits is in agreement (as evidenced in Fig. 4 of the main text). Furthermore, the diffusion coefficient parameter obtained from Lorentzian fits at 46 bar and 51.5 bar departs from the predicted hard sphere behavior more than the parameter extracted from the Gaussian fits (as evidenced in Supplementary Fig. 3). At the same time, employing Gaussian fits to the data at and above 67.5 bar leads to the opposite scenario.

- Lastly, our Bayesian analysis independently corroborates the gradual change from a Gaussian-like to a Lorentzian-like nature in the QENS spectra of methane with increasing pressure. This analysis unequivocally demonstrates that the Gaussian model is consistently preferable to the Lorentzian model for pressures below 46 bar, while the converse holds true for pressures exceeding 67 bar.

We trust that these clarifications adequately address the concerns raised and provide a more comprehensive understanding of our methodology and findings.

Concerning the value of experimental data vs simulations, in this specific contest: that one MD paper delivers very uncertain results means to me that, in absence of technical flaws, for that model system the crossover nature is unclear. Exactly as it

can happen for any experimental result on a specific system, like the one reported in this manuscript.

The Reviewer is correct in noting that the study by Pipich *et al.* yields highly uncertain results. However, this issue is not unique. There are numerous theoretical or molecular dynamics (MD) studies where contrasting conclusions regarding the location of the gas-like to liquid-like crossover emerge, depending on the specific potential or criterion employed for its definition. Examples include the works by D. Bolmatov *et al.* [Nature Commun. 4, 2331 (2013)], V. V. Brazhkin *et al.* [Phys. Rev. Lett. 111, 145901 (2013)], and V. V. Brazhkin *et al.* [Phys. Rev. E 85, 031203 (2012)].

This situation underlines the importance of experimental validation, as demonstrated in our work, which highlights the significance of the Widom line and the negligible role of the Frenkel line in delineating a gas-like to liquid-like crossover for the molecular diffusion within the supercritical state.

I reckon that the authors have softened some key statements and conclusiveness of their manuscript. This is good and points to the direction I indicated in my previous report. The manuscript could be published in a specialized journal, in such case the audience itself will have the instruments to judge the technical points leading to the claims, as well as the relevance of such claims within the context of the vast literature dealing with this topic.

In the previous revision, we have indeed amended the abstract and expanded the Introduction and Discussion sections to address comments of Reviewer #3. We are happy to see that the Reviewer recognizes this.

We disagree with the Reviewer's suggestion to publish the manuscript in a specialized journal. Given the broad interest in the topic of our paper and the ongoing debate within the scientific community, we believe that our work merits publication in Nature Communications. Our study identifies an unequivocal Gaussian-to-Lorentzian change in the dynamic structure factor of supercritical methane when crossing the Widom line, which has never been observed in any system, to the best of our knowledge, by either experimental or theoretical studies.

Regarding the role of the Widom line as the boundary between gas-like and liquid-like behaviours of supercritical systems, the present observation complements computational investigations published in journals such as Nature Physics and Nature Communications [G. G. Simeoni *et al.* Nature Phys. 6, 503-507 (2010); P. Gallo *et al.* Nature Commun. 5, 5806 (2014)] and demonstrates the negligible role of the Frenkel line in delineating the same crossover, to be compared with studies published for example in Physical Review Letters [V. V. Brazhkin *et al.* Phys. Rev. Lett. 111, 145901 (2013)] and (again) Nature Communications [D. Bolmatov *et al.* Nature Commun. 4, 2331 (2013)].

Reviewer #2 (Remarks to the Author):

I fully agree with the response and the opinion of the authors. I only would like to add some small comments only:

“First, we observe, as illustrated in Supplementary Fig. 1 and discussed in Supplementary Note 1, a clear crossing in energy of the spectral lines at constant Q within the pressure range of 46 – 67.5 bar, indicating an unequivocal change in spectral shape independent of any fitting model or procedure. If the spectral line remained the same, no crossing would be observed with changing pressure.”

[The spectra are normalized to their maximum (at E=0) on the figure, and in this case yes, otherwise there would be obviously a crossing upon line broadening.] It is an indication, true, but not a proof. E.g. in this energy range collective vibrations could also play a role. On the other hand, this observation is only an additional support. **The strongest evidence is undoubtedly provided by the Bayesian analysis, which is a clear mathematical, quantitative proof.** My last remark to this point is about the 4th argument, comparison with NMR. This argument is for me not complete, because it depends on the NMR data analysis. To my opinion, if you compare to NMR, it would be good to state how those values have been obtained. Depending on that, one can accept this as a proof or not. But even without this argument, the situation is clear, therefore this comment has no consequence on the manuscript.

Reading again the article, I find that the key observation is now stated clearly and fairly at the beginning of the manuscript, followed by a well founded, critical analysis and discussion. Furthermore, I believe the topic is of interest for a general audience and important enough for being published in Nat. Comm. Notably, conducting such experiments is far from trivial. The valuable experience and know-how of the authors in the field of high-pressure neutron scattering was crucial in this work. **I recommend publication of the manuscript in its present form.**

Fanni Juranyi

Below, the Reviewer's comments are shown in blue and our responses are in black (and in a different font type).

Reviewer #2 (Remarks to the Author):

I fully agree with the response and the opinion of the authors. I only would like to add some small comments only:

“First, we observe, as illustrated in Supplementary Fig. 1 and discussed in Supplementary Note 1, a clear crossing in energy of the spectral lines at constant Q within the pressure range of 46 – 67.5 bar, indicating an unequivocal change in spectral shape independent of any fitting model or procedure. If the spectral line remained the same, no crossing would be observed with changing pressure.”

[The spectra are normalized to their maximum (at $E=0$) on the figure, and in this case yes, otherwise there would be obviously a crossing upon line broadening.] It is an indication, true, but not a proof. E.g. in this energy range collective vibrations could also play a role. On the other hand, this observation is only an additional support. The strongest evidence is undoubtedly provided by the Bayesian analysis, which is a clear mathematical, quantitative proof. My last remark to this point is about the 4th argument, comparison with NMR. This argument is for me not complete, because it depends on the NMR data analysis. To my opinion, if you compare to NMR, it would be good to state how those values have been obtained. Depending on that, one can accept this as a proof or not. But even without this argument, the situation is clear, therefore this comment has no consequence on the manuscript.

Reading again the article, I find that the key observation is now stated clearly and fairly at the beginning of the manuscript, followed by a well founded, critical analysis and discussion. Furthermore, I believe the topic is of interest for a general audience and important enough for being published in Nat. Comm. Notably, conducting such experiments is far from trivial. The valuable experience and knowhow of the authors in the field of high-pressure neutron scattering was crucial in this work. I recommend publication of the manuscript in its present form.

We thank again the Reviewer for recommending publication.

We agree with the Reviewer's comments: the Bayesian analysis is certainly the most quantitative proof. The crossing of the normalized spectra is indeed an indication but not a proof. The comparison with literature NMR results is an additional strong indication. It is true that the NMR data analysis plays a role; however, in this case, the self-diffusion coefficient values obtained by NMR are reproduced by various sets of simulations for the hard-sphere fluid and also by simulations of real methane (for the limited number of pressure/temperature conditions at which such simulations exist in the literature for methane).